# Multi-Scale Digital Pathology Patch-Level Prostate Cancer Grading Using Deep Learning: Use Case Evaluation of DiagSet Dataset

**DOI:** 10.3390/bioengineering11060624

**Published:** 2024-06-18

**Authors:** Tanaya Kondejkar, Salah Mohammed Awad Al-Heejawi, Anne Breggia, Bilal Ahmad, Robert Christman, Stephen T. Ryan, Saeed Amal

**Affiliations:** 1College of Engineering, Northeastern University, Boston, MA 02115, USA; kondejkar.t@northeastern.edu (T.K.); s.al-heejawi@northeastern.edu (S.M.A.A.-H.); 2MaineHealth Institute for Research, Scarborough, ME 04074, USA; anne.breggia@mainehealth.org; 3Maine Medical Center, Portland, ME 04102, USA; bilal.ahmad@spectrumhcp.com (B.A.); robert.christman@spectrumhcp.com (R.C.); stephen.ryan@mainehealth.org (S.T.R.); 4The Roux Institute, Department of Bioengineering, College of Engineering, Northeastern University, Boston, MA 02115, USA

**Keywords:** machine learning, prostate cancer classification, health care

## Abstract

Prostate cancer remains a prevalent health concern, emphasizing the critical need for early diagnosis and precise treatment strategies to mitigate mortality rates. The accurate prediction of cancer grade is paramount for timely interventions. This paper introduces an approach to prostate cancer grading, framing it as a classification problem. Leveraging ResNet models on multi-scale patch-level digital pathology and the Diagset dataset, the proposed method demonstrates notable success, achieving an accuracy of 0.999 in identifying clinically significant prostate cancer. The study contributes to the evolving landscape of cancer diagnostics, offering a promising avenue for improved grading accuracy and, consequently, more effective treatment planning. By integrating innovative deep learning techniques with comprehensive datasets, our approach represents a step forward in the pursuit of personalized and targeted cancer care.

## 1. Introduction

Prostate cancer stands as a major health concern globally, necessitating precise diagnostic and grading methodologies for effective therapeutic interventions. Prostate cancer is characterized by abnormal cell growth in the prostate gland. It is the second most common cancer in the world and the most common cancer in men. Each year, over 1.4 million new cases of prostate cancer are found, causing more than 375,000 deaths [1]. It typically starts in the cells of the prostate gland and can grow slowly, initially confined to the gland itself, or it may spread rapidly to other parts of the body if left untreated. Early detection through screening tests such as prostate-specific antigen (PSA) tests and digital rectal exams (DREs) is crucial for timely treatment and improved outcomes. The processing and scanning of tissue biopsies in the laboratory produce large whole-slide images (WSIs), enhancing workflow efficiency and reproducibility. One characteristic of PCa is its tendency to be “non-aggressive”, which can complicate treatment decisions and determine the necessity of more serious interventions. To address this challenge, the Gleason Grading System classifies tumors into numerical risk groups known as WHO/ISUP grade groups, as adopted by the International Society of Urological Pathology (ISUP) and the World Health Organization (WHO). The conventional grading of prostate cancer involves challenges marked by intra- and inter-observer variability, potentially leading to suboptimal treatment decisions. Advancements in imaging techniques and molecular biomarkers offer promising avenues for enhancing the precision of prostate cancer diagnosis and risk stratification, guiding personalized treatment strategies tailored to individual patients’ needs. Integrating these innovative approaches into clinical practice holds the potential to improve prognostic accuracy and optimize therapeutic outcomes for men diagnosed with prostate cancer. To address these complexities, the integration of machine learning (ML) and deep learning (DL) techniques has emerged as a promising frontier, revolutionizing the landscape of prostate cancer diagnosis and grading. These technologies can enhance the accuracy of diagnostic processes, reducing the variability associated with human interpretation. This, in turn, can lead to more reliable and personalized treatment decisions for individuals diagnosed with prostate cancer. One essential aspect of this diagnostic is the Gleason score grading system. The Gleason score provides crucial insights into the aggressiveness of prostate cancer by evaluating the patterns observed in cancerous tissue under a microscope. Higher Gleason scores correlate with more aggressive cancer, aiding clinicians in determining optimal treatment strategies tailored to individual patients. By integrating the Gleason score into diagnostic and treatment protocols, healthcare professionals can enhance precision and personalized care for individuals with prostate cancer. This integration not only facilitates more accurate grading but also allows for the development of predictive models that assist clinicians in tailoring treatment plans based on individual patient characteristics. By leveraging advanced computational methods, machine learning models can analyze vast amounts of medical data, including imaging and pathology reports, to identify subtle patterns and markers indicative of the severity of prostate cancer. This approach not only assists in more accurate grading but also allows for the development of predictive models that can help healthcare professionals tailor treatment plans based on individual patient characteristics. In this research paper, we perform a comparative study of ResNet models within the domain of prostate cancer diagnosis and grading. Our primary objective is to surpass the accuracies achieved in the previous literature by leveraging the advancements in deep learning techniques. Through our research, we wish to contribute to the ongoing efforts in revolutionizing the landscape of prostate cancer diagnosis and grading, with the goal of enhancing patient outcomes and personalized care. Recent advancements in artificial intelligence (AI) and machine learning (ML) have reshaped prostate cancer diagnosis and Gleason grading. Goldenberg et al. [2] provide an overview of AI and ML’s potential in prostate cancer management, showing performance comparable to traditional diagnostic methods. Abraham and Nair [3] developed a deep learning algorithm for histopathologic diagnosis and Gleason grading, focusing on the PROSTATEx-2 2017 dataset. A study in *The Lancet Oncology* also presents an AI system for prostate biopsies, demonstrating accuracy and performance similar to expert pathologists [4]. SVM-based analysis in the work by Bhattacharjee et al. [5] enriches the AI landscape, accurately classifying Gleason grading based on biopsy-derived image features. These studies highlight the promising trajectory of AI and ML applications, offering accurate and efficient methodologies for prostate cancer grading. Goldenberg et al. [2] explored ML and deep learning (DL) techniques such as support vector machines (SVMs) and convolutional neural networks (CNNs) for diagnostic imaging in prostate cancer; using the PROSTATEx challenge dataset, the study shows performance comparable to radiologists. In *The Lancet Oncology* [6], a study focuses on an AI system for prostate biopsies, achieving remarkable performance in distinguishing benign and malignant biopsy cores. Abraham and Nair [4] introduced a novel approach for grading prostate cancer using a VGG-16 convolutional neural network and an ordinal class classifier, contributing to histopathologic diagnosis and Gleason grading. Bhattacharjee et al. [5] presented a quantitative analysis of benign and malignant tumors in histopathology, predicting prostate cancer grading using SVM based on biopsy-derived images. The collective findings from these studies demonstrate AI and ML’s transformative potential in prostate cancer diagnosis and grading. In 2022, Kamal Hammouda et al. [7]. conducted a study at Radboud University Medical Center involving 3080 whole-slide images (WSIs). The study focused on multi-level binary classification into Gleason grades using a multi-stage classification-based deep learning approach. Their findings demonstrated an overall accuracy of 66.23%, showcasing the potential of deep learning methods in prostate cancer grading. These methods offer comparable performance to traditional approaches while presenting opportunities for automation, reducing workload, and providing expertise in resource-limited regions. In 2024, Wang et al. explored a novel method for diagnosing multiple types of cancer using an environmentally friendly and cost-effective approach [8]. This method addresses the need for accessible diagnostic tools, especially in remote or resource-limited regions, by using dried serum spots instead of traditional liquid blood storage. This technique not only reduces the environmental impact but also ensures the stability of metabolites. Table 1 summarizes the key findings in each paper.

## 2. Proposed Methods

The primary objective of our proposed work is to enhance the accuracy of prostate cancer diagnosis and Gleason grading using advanced machine learning techniques [9]. It aims to implement Resnet models and compare their accuracies for the better classification and grading of prostate cancer [10]. It also proposes a model which uses the segmentation model as a feature extractor for the CNN models trained. Our proposed work aims to advance prostate cancer diagnosis and Gleason grading through the utilization of the histopathological dataset, DiagSet. This dataset, comprising over 2.6 million tissue patches from 430 fully annotated scans, 4675 scans with binary diagnoses, and 46 scans independently diagnosed by histopathologists, provides a rich resource for in-depth analysis. It includes meticulously annotated tissue patches, binary diagnoses, and expert-assigned diagnoses (Accessible at https://github.com/michalkoziarski/DiagSet (accessed on 3 June 2024)) It provides a foundation for our research, ensuring a thorough investigation into the factors influencing model performance. Our approach is centered around ensembles of convolution neural networks operating on histopathological scans at different scales [11]. We implement a CNN framework tailored to the characteristics of DiagSet [12]. This involves the detection of cancerous tissue regions and the prediction of scan-level diagnosis [13,14].

### Dataset

Each whole-slide image is systematically partitioned into 256 × 256 blocks, forming the basis for a detailed analysis using convolutional neural network (CNN) classifiers [13]. These classifiers categorize each block into one of nine distinct classes (Figure 1): scan background (BG); tissue background (T); normal, healthy tissue (N); acquisition artifact (A); or one of the five Gleason grades (R1R5) [14]. Leveraging the versatility of pre-trained ResNet models, specifically ResNet-18, ResNet-34, and ResNet-50, our methodology ensures a comprehensive examination of the dataset at varying levels of complexity [15,16]. This multi-model approach enhances the robustness of our analysis, capturing intricate features within each block and facilitating a nuanced understanding of prostate cancer pathology across different Gleason grades and tissue types [17,18,19,20]. Furthermore, this extends across multiple magnification levels, including 40×, 20×, 10×, and 5×. At each magnification, the systematic division and classification of 256 × 256 blocks are repeated, allowing for a comprehensive analysis of prostate cancer pathology at varying resolutions. By comparing the accuracies obtained at different magnifications, our approach aims to discern patterns and variations in classification performance, providing insights into the robustness and scalability of the proposed model across a spectrum of image resolutions. This multi-level evaluation enhances the reliability and generalizability of our findings, acknowledging the significance of adapting to diverse magnification contexts in the field of histopathological analysis.

## 3. Methodology

At each magnification level (40×, 20×, 10×, and 5×), structured methodology was implemented to prepare and assess the dataset with thoroughness and precision.

### 3.1. Dataset

Initially, the dataset underwent a partitioning process, meticulously dividing it into distinct training and testing sets to ensure an equitable distribution of images across various classes. This step was crucial in mitigating potential biases and ensuring the representativeness of the data. Moreover, to address potential class imbalances and enhance the efficacy of model training, a carefully curated subset consisting of 4000 images per class was meticulously selected. This deliberate curation aimed to provide the model with a diverse array of examples, thereby fortifying its ability to generalize proficiently across a wide spectrum of scenarios. By preparing the dataset in this manner, we aimed to ensure that our model could effectively learn and generalize from a comprehensive and representative set of histopathological images. This approach not only improved the model’s performance but also bolstered its robustness and reliability in real-world applications of prostate cancer diagnosis and Gleason grading.

### 3.2. Training Phase

The selection of ResNet models ResNet-18, ResNet-34, and ResNet-50 was based on their established effectiveness in capturing intricate features relevant to various Gleason grades and tissue types across various levels of magnification [21,22,23,24,25,26,27,28,29,30,31,32,33,34,35,36,37,38]. The decision to employ these architectures was based on their proven capabilities to handle the complexities of histopathological images effectively. We utilized transfer learning to enhance the accuracy and robustness of our prostate cancer grading models. By leveraging the pre-existing learned features from the initial training on general image datasets, the models could better capture intricate patterns and details specific to histopathological images of prostate cancer [38,39,40]. The fine-tuning process involved freezing the initial layers of the ResNet models to retain the general features learned and only training the later layers to adapt to the specific characteristics of the dataset. This approach improved the convergence speed and enhanced the models’ overall accuracy by focusing on the relevant medical imaging features unique to prostate cancer pathology. The training phase spanned 100 epochs and employed pre-trained ResNet models, including ResNet18, ResNet-34, and ResNet-50 architectures. Each ResNet model underwent extensive training on the curated dataset, with the objective of fine-tuning their parameters to accurately classify tissue patches into one of the nine distinct classes, including scan background; tissue background; normal, healthy tissue; acquisition artifact; and the five Gleason grades. During the training phase, a 5-fold cross-validation approach was used to enhance the model’s robustness [40]. This involved partitioning the dataset into five subsets, with each subset used as a validation set once while the remaining four subsets were used for training. This process was repeated five times, with each subset used exactly once as the validation set. Implementing 5-fold cross-validation helped to prevent overfitting by systematically dividing the dataset into multiple subsets for training and validation, ensuring that the model’s performance was robust and reliable on unseen data. It enhanced the model’s ability to generalize well to new, unseen data by evaluating its performance across diverse scenarios. Additionally, cross-validation mitigates biases or inconsistencies in the dataset, leading to a more stable and resilient model [41,42,43,44]. It also facilitates the optimization of model hyperparameters and provides more reliable evaluation metrics, instilling confidence in the model’s predictions. By iteratively training and validating the model on different subsets of the data, the 5-fold cross-validation technique helped ensure that the model’s performance was not overly dependent on any subset of the data, thus improving its generalization ability [45,46,47,48].

### 3.3. Testing Phase

For the testing phase, a stratified split function was employed to maintain class balance in the testing dataset; 7200 images were used. This strategic approach ensured a thorough evaluation of the model’s performance across diverse classes and magnification levels. Stratification was imperative to prevent any bias in the evaluation process and to guarantee that the model’s performance was assessed comprehensively across all classes [49]. The incorporation of pre-trained ResNet models, coupled with meticulous data preparation and testing strategies, underscores the resilience and adaptability of the proposed methodology [50,51,52]. This comprehensive approach contributes to the accuracy of prostate cancer diagnosis and Gleason grading, establishing a robust foundation for reliable histopathological analysis across different magnification contexts. In the next stage of model development, we integrated the DeepLabv3 segmentation model to augment the feature extraction process, thus enhancing the predictive capabilities of our classification models [53]. This integration is designed to strengthen the model’s reliability and accuracy. The process involves feeding input images to DeepLabv3, which is optimized to efficiently extract relevant features. The output produced by the segmentation model, which encapsulates detailed spatial and semantic information, is directed to a convolutional neural network (CNN) classifier. This CNN classifier utilizes the extracted features to generate predictions, leveraging the understanding provided by the DeepLabv3 segmentation model. Through this comprehensive approach, we aim to improve the accuracy and robustness of our predictive model, advancing its performance in classification tasks. The initialization of both the ResNet and DeepLabv3 models serves as the foundational step in this process. Parameter freezing is implemented to safeguard the integrity of both models during training, except for the final fully connected layer of ResNet models, which undergoes modification to accommodate the classification requirements of the dataset. By preserving the feature extraction capabilities of DeepLabv3 and subsequently fine-tuning the ResNet models using these extracted features, the combined model aims to harness the collective strengths of both models to achieve enhanced performance and generalization on the target classification task. This approach capitalizes on the transfer learning potential of ResNet and the feature-rich representations generated by DeepLabv3, thereby facilitating more accurate and robust predictions, particularly in scenarios with limited labeled data availability. The DeepLabv3 model is specifically used to improve the feature extraction process by capturing detailed spatial and semantic information from the histopathological images. This information is crucial for distinguishing between different tissue types and Gleason grades. This innovative approach offers several distinct advantages. Firstly, by leveraging the feature extraction capabilities of DeepLabv3, the model aptly captures the intricate and nuanced features inherent in the input images. This facilitates a richer representation of the data, enabling the CNN classifier to make more precise and accurate predictions [54]. Furthermore, the integration of a segmentation model as a feature extractor allows our model to discern spatial relationships and semantic context within the images. Additionally, the segmentation model helps in isolating the relevant regions of interest within the images, reducing noise and irrelevant data that could potentially hinder the classification performance [55].

## 4. Results

In this study, we conducted a comprehensive evaluation of our proposed model’s performance across various magnification levels (40×, 20×, 10×, and 5×) in the context of prostate cancer diagnosis and Gleason grading. Leveraging ResNet18, ResNet34, and ResNet50 architectures, our model highlighted remarkable adaptability and accuracy throughout the investigation. For ResNet18, testing accuracies ranged from 0.9956 at 10× magnification to 0.9992 at 20× magnification, indicating a high level of reliability in predictions across different image resolutions. The model’s performance was slightly lower at the 10× magnification, suggesting potential challenges in capturing finer details at this scale. This could be attributed to the relatively simpler architecture of ResNet18, which may not be as effective in identifying complex tissue patterns at lower magnifications. ResNet34 exhibited even higher testing accuracies, consistently achieving 0.9999 at multiple magnifications (40× and 20×) and only slightly lower accuracies at other magnifications (0.9957 at 10× and 0.9993 at 5×), showcasing its robust ability to handle a variety of inputs with exceptional accuracy. The higher number of layers in ResNet34 allows for better feature extraction, especially at higher magnifications where detailed tissue structures are more prominent. ResNet50 also performed impressively, achieving testing accuracies between 0.9915 at 10× magnification and 0.9981 at 5×, demonstrating its competence in maintaining high levels of performance across the varying levels of image magnifications. The deeper layers in ResNet50 likely contribute to its ability to generalize well across different resolutions, but the slight dip at 10× suggests that the added complexity may sometimes lead to overfitting on finer details. The graphs in Figure 2 show training accuracy, validation accuracy in training, and validation loss for Resnet34 model on the 20× magnification image set.

The following graphs show training loss for each fold, accuracies, and validation loss for the Resnet34 model on a 20×-magnification image set when 5-fold cross-validation is used. In the graph of training accuracy, the third and fourth folds show higher accuracies and extremely low losses from the initial epochs. A possible explanation is that the images in those folds are similar to the ones used while training the model. By evaluating the model’s performance under both scenarios, we were able to quantify the tangible benefits of employing the 5-fold cross-validation technique. The graphical representations above in Figure 2 and Figure 3 offer a clear understanding of the model’s learning trajectory and its performance dynamics during training and validation stages. The Appendix A show learning losses and accuracies for all Resnet architectures across all magnifications.

Table 2 shows the model’s robustness and generalization ability, highlighting improvements achieved through the validation process. Figure 3 depicts the training loss for each fold over a group of epochs (one set represents 15 epochs), along with accuracies and validation losses for the ResNet34 model on 20× magnification. The losses for each fold are different because despite having a balanced dataset, the stratification process during the creation of folds can still result in slight variations in class distribution within each fold, impacting the loss differently. Certain folds might contain more challenging samples, which can significantly affect the loss curves for those specific folds. 

The performance differences across magnification levels and ResNet models correlate with the complexity of tissue patterns in the dataset. At higher magnifications (40× and 20×), detailed tissue structures are more pronounced, allowing deeper models like ResNet34 and ResNet50 to leverage their complex architectures for better feature extraction.

These findings collectively emphasize the adaptability, reliability, and versatility of ResNet18, ResNet34, and ResNet50 in prostate cancer diagnosis and Gleason grading across different magnification levels. The ability of these models to maintain high accuracy rates across varying resolutions is essential for ensuring accurate and consistent diagnostic assessments, thereby contributing to improved patient outcomes and clinical decision-making in the field of prostate cancer pathology.

## 5. Discussion

This study demonstrates the effectiveness of using ResNet models like ResNet18, ResNet34, and ResNet50 in diagnosing prostate cancer and grading Gleason scores. These models perform admirably across various magnification levels, indicating their proficiency in detecting crucial details in histopathological images. However, there is potential for further exploration into other algorithms. Exploring techniques like Vision Transformers could potentially improve the models’ ability to discern subtle variations, leading to enhanced diagnostic accuracy. Moreover, the success of the proposed approach in prostate cancer diagnosis suggests its potential applicability to other malignancies. Expanding the methodology to include datasets and histopathological images from different cancer types could facilitate more accurate diagnosis and grading across various cancers. This multi-cancer approach could contribute to a broader understanding of histopathological features and aid in the development of comprehensive diagnostic tools applicable across diverse oncological contexts. Moving forward, an intriguing prospect involves transforming these developed models into an accessible web application. This would democratize access to advanced prostate cancer diagnostic tools, enabling healthcare professionals to integrate machine learning into their clinical practice more seamlessly. The application could offer features such as the real-time analysis of histopathological images, automated Gleason grading, and seamless integration with existing electronic medical record systems for streamlined patient management. An important consideration for future work includes the potential expansion of modalities in electronic health record (EHR) data to further enhance the model’s capabilities. Integrating additional types of EHR data, such as radiological imaging, genetic information, and laboratory test results, could provide a more holistic approach to diagnosing and grading prostate cancer. This comprehensive utilization of patient data can improve model accuracy, leading to better patient outcomes and personalized treatment plans.

## 6. Conclusions

The proposed approach, framed as a classification problem and utilizing ResNet models in conjunction with the Diagset dataset, highlights its potential as a robust tool for prostate cancer diagnosis and Gleason grading. The integration of machine learning and deep learning techniques represents a pivotal advancement in prostate cancer diagnosis and grading. The proposed methodology, centered around convolutional neural networks operating on histopathological scans at multiple magnification levels, presents a comprehensive and robust approach to prostate cancer diagnosis and Gleason grading. Leveraging the rich resource provided by the DiagSet dataset, this research offers insights into the intricate features underlying prostate cancer pathology across different Gleason grades and tissue types. The comprehensive evaluation of the proposed model’s performance across various magnification levels reaffirms its adaptability and reliability in accurately assessing prostate cancer pathology. The consistent high accuracies achieved by the ResNet models further validate their stability and efficacy, positioning them as valuable assets in clinical practice for prostate cancer diagnosis and Gleason grading. Moving forward, an intriguing prospect involves transforming these models into an accessible web application, which would democratize access to advanced diagnostic tools. This application would allow healthcare professionals to seamlessly integrate machine learning into clinical practice, offering features such as the real-time analysis of histopathological images and automated Gleason grading. Additionally, the application would facilitate integration with electronic medical record systems, enabling streamlined patient management and improved clinical workflows. By providing an intuitive interface, the web application would ensure that sophisticated machine learning tools are accessible to clinicians without requiring extensive technical expertise, thereby enhancing diagnostic accuracy and efficiency in diverse healthcare settings.

This study establishes a solid foundation for advancing prostate cancer diagnosis and Gleason grading through machine learning. Future research should explore additional algorithms, such as Vision Transformers, and improve model efficiency to further enhance accuracy, scalability, and clinical utility. By addressing these areas, future work can expand the capabilities and applications of these models, ultimately benefiting healthcare providers and patients alike.

## Figures and Tables

**Figure 1 bioengineering-11-00624-f001:**
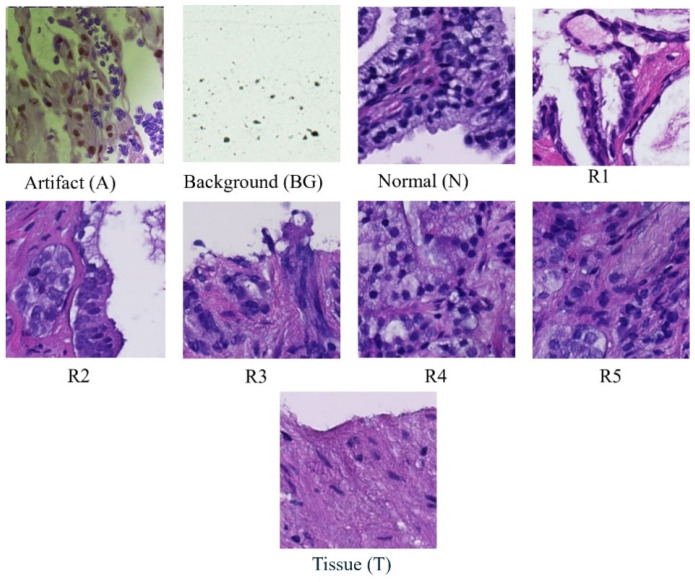
Samples of classes in dataset.

**Figure 2 bioengineering-11-00624-f002:**
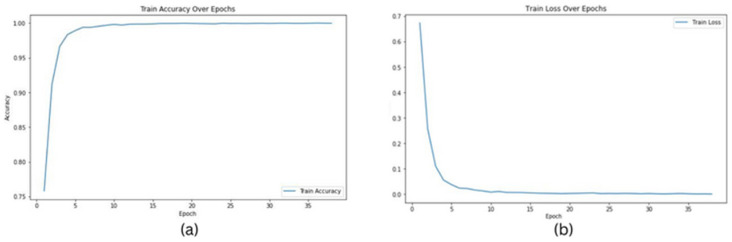
Graphs for Resnet34 model on images of 20× magnification: (**a**) training accuracy, (**b**) training loss.

**Figure 3 bioengineering-11-00624-f003:**
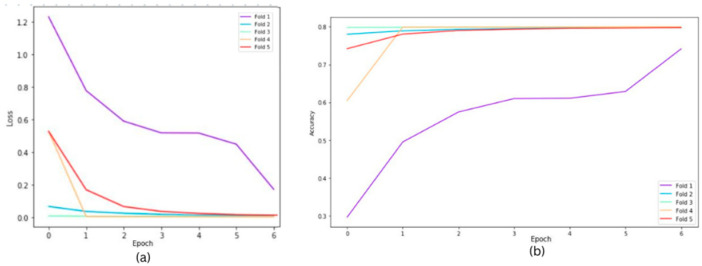
Cross fold training loss (**a**) and training accuracies (**b**) for ResNet34 model on 20× magnification.

**Table 1 bioengineering-11-00624-t001:** Summary of methodologies and datasets used in prostate cancer studies [2,3,4,5].

Paper	Journal/Conference	Author + Year	Methodology	Dataset
1.	A new era: artificial intelligence and machine learning in prostate cancer (Nature Reviews Urology) [2]	Goldenberg, S.L.; Nir, G.; Salcudean, S.E. 2019	ML and DL techniques for diagnostic imaging, SVM, CNN-based DL network	PROSTATE-x challenge, mpMRI images
2.	Automated grading of prostate cancer using CNN and ordinal class classifier (Informatics in Medicine Unlocked) [3]	Abraham, B.; Nair, M.S. 2019	VGG-16 CNN, Ordinal Class Classifier with J48 Achieved a moderate quadratic weighted kappa score of 0.4727 in grading PCA into 5 grade groups. Positive predictive value of 0.9079 in predicting clinically significant prostate cancer.	PROSTATEx-2 2017 grand challenge dataset
3.	AI for diagnosis and grading of prostate cancer in biopsies (The Lancet Oncology) [4]	Ström, P.; Kartasalo, K.; Olsson, H.; Solorzano, L.; Delahunt, B.; Berney, D.M.; Bostwick, D.G.; Evans, A.J.; Grignon, D.J.; Humphrey, P.A.; et al. 2020	Deep neural networks for biopsy assessment. AI system achieved high accuracy in distinguishing benign and malignant biopsy cores (AUC of 0.997 and 0.986 on respective datasets).	STHLM3 diagnostic study, external validation dataset
4.	Quantitative Analysis of Benign and Malignant Tumors in Histopathology: Predicting Prostate Cancer Grading Using SVM (Applied Sciences) [5]	Bhattacharjee, S.; Park, H.-G.; Kim, C.-H.; Prakash, D.; Madusanka, N.; So, J.-H.; Cho, N.-H.; Choi, H.-K. 2019	SVM classification, Image manipulation, K-means, Watershed algorithms. Accuracy of 88.7% for malignant vs. benign, 85.0% for Grade 3 vs. Grade 4, 5, and 92.5% for Grade 4 vs. Grade 5.	Biopsy-derived images, Gleason grade groups (Grade 3, Grade 4, Grade 5, and benign)

**Table 2 bioengineering-11-00624-t002:** Models’ performance on 5-fold cross-validation.

Magnification	Architecture	Average Training Accuracy	Testing Accuracy
40×	ResNet18	0.9995	0.9977
20×		0.9996	0.9992
10×		0.9993	0.9964
5×		0.9995	0.9921
40×	ResNet34	0.9992	0.9999
20×		0.9993	0.9999
10×		0.9998	1.0000
5×		0.9993	0.9993
40×	ResNet50	0.9993	0.9957
20×		0.9991	0.9915
10×		0.9956	0.9952
5×		0.9893	0.9981

## Data Availability

The used datasets are publicly available as specified in their references.

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
