# Peer review of "Multi-Scale Digital Pathology Patch-Level Prostate Cancer Grading Using Deep Learning: Use Case Evaluation of DiagSet Dataset"

_bioengineering, 2024, doi:10.3390/bioengineering11060624_

Round 1

Reviewer 1 Report

Comments and Suggestions for Authors

The methodological innovation is limited. The authors used routine methods. The qualitative results should also be provided. The analysis part is week, the quality of figures is very poor.

Author Response

Dear Reviewer,

We would like to express our deepest gratitude for your time and expertise in reviewing our manuscript. Your insightful comments have been important in refining our research article. We have carefully considered your suggestions and made corresponding revisions to the manuscript. Thank you once again for your invaluable contribution.

Regards

Reviewer 2 Report

Comments and Suggestions for Authors

The manuscript introduces an approach to prostate cancer grading, framing it as a classification problem. Leveraging ResNet models on multi-scale patch-level digital pathology and the Diagset dataset, the proposed method achieved an accuracy of 0.999 in identifying clinically significant prostate cancer. The study contributes to the evolving landscape of cancer diagnostics.

1.     Please describe criteria in image selection from the dataset. The author claimed that they meticulously selected 4000 images per class.

2.     How many images were used in training phase? And how many images were used in test phase?

3.     In figure 2, why the loss and accuracy in Fold 1 were so different with the rest of folds?

4.     More related advances on machine learning and cancer diagnosis (VIEW 2023, 4, 20220038; Nature Sustainability 2024, 10.1038/s41893-024-01323-9) should be included and discussed.

5.     The names of the X-axis in Figure 1b, Figure 2 were missed.

Comments on the Quality of English Language

/

Author Response

(The authors gave the same response as above.)

Round 2

Reviewer 1 Report

Comments and Suggestions for Authors

The integration of DeepLabv3 segmentation for feature extraction is a key strength of your study. However, the manuscript lacks detailed explanation on how this integration specifically enhances model performance.

While your results are impressive, a direct comparison with existing methods for prostate cancer grading is limited.

he results section provides accuracy metrics for different magnification levels and ResNet models but lacks an in-depth analysis.

The following literature will further enrich the introduction and enhance the quality of the manuscript.

1 - Association between shift works and and risk of prostrate cancer: a systematic review and meta-analysis of observational studies: carcinogenesis

2 - 68Ga-PSMA-11 PET/CT versus 68Ga-PSMA-11 PET/MRI for the detection of biochemically recurrent prostate cancer: a systematic review and meta-analysis. FRONTIERS IN ONCOLOGY

3 - A Dual-domain Diffusion Model for Sparse-view CT Reconstruction. IEEE Signal Processing Letters. doi: 10.1109/LSP.2024.3392690

Author Response

Dear Reviewer,

We would like to express our deepest gratitude for your time and expertise in reviewing our manuscript. Your insightful comments have been important in refining our research article. We have carefully considered your suggestions and made corresponding revisions to the manuscript. Thank you once again for your invaluable contribution.

Regards,

Dr. Salah Alheejawi
